# Serum Free Light-Chain Ratio at Diagnosis Is Associated with Early Renal Damage in Multiple Myeloma: A Case Series Real-World Study

**DOI:** 10.3390/biomedicines10071657

**Published:** 2022-07-10

**Authors:** Danilo De Novellis, Raffaele Fontana, Angela Carobene, Bianca Serio, Idalucia Ferrara, Maria Carmen Martorelli, Laura Mettivier, Roberto Guariglia, Serena Luponio, Immacolata Ruggiero, Matteo D’Addona, Tiziana Di Leo, Valentina Giudice, Carmine Selleri

**Affiliations:** 1Hematology and Transplant Center, University Hospital “San Giovanni di Dio e Ruggi d’Aragona”, 84131 Salerno, Italy; danilo.denovellis@sangiovannieruggi.it (D.D.N.); raffaele.fontana@sangiovannieruggi.it (R.F.); angela.carobene@studenti.unina.it (A.C.); bianca.serio@sangiovannieruggi.it (B.S.); idalucia.ferrara@sangiovannieruggi.it (I.F.); mc.martorelli@sangiovannieruggi.it (M.C.M.); laura.mettivier@sangiovannieruggi.it (L.M.); roberto.guariglia@sangiovannieruggi.it (R.G.); serena.luponio@sangiovannieruggi.it (S.L.); i.ruggiero@studenti.unisa.it (I.R.); matteo.daddona@studenti.unina.it (M.D.); cselleri@unisa.it (C.S.); 2Clinical Pathology and Biochemistry Unit, University Hospital “San Giovanni di Dio e Ruggi d’Aragona”, 84131 Salerno, Italy; tiziana.dileo@sangiovannieruggi.it; 3Department of Medicine, Surgery, and Dentistry, University of Salerno, 84081 Baronissi, Italy

**Keywords:** multiple myeloma, free light-chain ratio, renal failure, prognosis, free light chains

## Abstract

The serum free light-chain (FLC) ratio is a sensitive tool for the differential diagnosis of plasma cell disorders and is biomarker of multiple myeloma (MM) progression from premalignant conditions. Here, we investigate the potential role of FLC ratio at diagnosis in identifying early renal damage in MM patients and other correlations with clinical, laboratory, and molecular findings. A total of 34 MM patients who had undergone autologous stem cell transplantation were included in this retrospective case series study, and FLC quantification was performed with nephelometric assays. In our study, sFLC ratio was significantly associated with light-chain MM and β-2 microglobulin levels, likely indicating a high disease burden at diagnosis, especially in patients without heavy chain M-protein at serum electrophoresis. Moreover, the sFLC ratio was inversely correlated with glomerular filtration rate, possibly identifying early renal damage in MM patients. Our preliminary results confirm the importance of early sFLC evaluation, especially in patients with the light-chain MM type and low disease burden, to minimize the risk of late renal failure.

## 1. Introduction

Multiple myeloma (MM) is a clonal hematological disorder characterized by the expansion of malignant plasma cells producing monoclonal paraprotein in the bone marrow (BM) [1,2]. MM accounts for 1% of all cancers and 10% of hematological malignancies, with a yearly age-adjusted incidence in the United States of 4 per 100,000 [3] and a median age at diagnosis of 70 years [4]. Malignant clones derive from neoplastic transformation of long-lived post-germinal center plasma cells in which genetic and epigenetic alterations occur during isotype class switching and somatic hypermutation [1,5,6,7]. The increased expression of MYC or the presence of *KRAS* mutations can also favor the transition from premalignant forms into a symptomatic and aggressive MM [5].

The main clinical feature of MM is the hyperproduction of serum and urine mono-clonal heavy and light-chain paraproteins (M-proteins) that accumulate in tissues, ultimately leading to diffuse organ damage, especially renal failure [2]. The most frequently found isotype of monoclonal immunoglobulins (Ig) in MM patients is IgG, accounted for 52% of all cases, followed by IgA, described in 21% of subjects, and light-chain-only secretion in 16%, while IgD and IgM phenotypes are rare (2% and 0.5% of cases, respectively) [8]. On the basis of recent International Myeloma Working Group (IMWG) recommendations, MM diagnosis requires the presence of at least one CRAB criterion among hypercalcemia, renal failure, anemia, and bone lesions, and myeloma defining events (MDEs), which are biomarkers of malignancy and progression in the absence of CRAB criteria, including BM clonal plasma cells ≥60%, magnetic resonance imaging (MRI) with at least one focal lesion >5 mm, and serum free light-chain (FLC) ratio ≥100 [9].

Serum FLC (sFLC) ratio detection by nephelometry or immunoassay is now considered to be an important biomarker of disease progression from monoclonal gammopathy of undetermined significance (MGUS), the premalignant MM form, to smoldering MM or to extended organ damage [2,9,10,11,12]. Free kappa or lambda chain hyperproduction is an indirect index of clonal plasma cell expansion and proliferation [13]; therefore, serum FLC ratio is an optimal minimally invasive biomarker of the differential diagnosis of benign and malignant plasma cell disorders [14,15]. Indeed, sFLC is a breakthrough biomarker of MM with a sensitivity close to 100% for non-secretory myeloma, as its FLC ratio is always abnormal, even in the absence of M-protein, thus rendering this marker an optimal tool for monitoring oligo- or non-secretory plasma cell disorders [16,17,18,19]. Moreover, sFLC is used for screening together with protein electrophoresis, immunofixation, and 24 h urine studies [17].

Therefore, plasma cell disorders are heterogeneous in clinical presentation and outcomes, and in molecular and immunophenotypic features, also influencing therapeutic strategy choices [20,21]. However, better risk stratification and prognostic definition are required to improve the clinical management and quality of life of those patients; thus, the discovery of new biomarkers is essential for understanding disease biology, and for identifying new possible therapeutic targets. In this retrospective case series study, we evaluated the influence of FLC ratio on clinical and laboratory features of MM patients at diagnosis and on outcomes.

## 2. Materials and Methods

### 2.1. Study Cohort

In this single-center retrospective observational study, we reviewed the medical records of fit patients with newly diagnosed MM who had undergone autologous hematopoietic stem cell transplantation (auto-HSCT) followed at the Hematology and Transplant Centre, University Hospital San Giovanni di Dio e Ruggi d’Aragona, Salerno, Italy, from January 2015 to October 2021, and a total of 34 subjects were included in our analysis. Patients received diagnosis of MM and chemotherapy as per the international guidelines after informed consent had been obtained in accordance with Declaration of Helsinki [22], and protocols approved by the local ethic committee (Campania Sud; prot./SCCE n. 24988). The included patients: were aged >18 years old, were diagnosed with MM according to 2014 IMWG criteria [2,23], and received chemotherapy followed by auto-HSCT. Clinical characteristics are summarized in Table 1.

Patients received first-line treatment as per the international protocols (Table 2): PAD, bortezomib 1.3 mg/m^2^, dexamethasone 40 mg daily, and doxorubicin 9 mg/m^2^ (*n* = 2); VRD, bortezomib 1.3 mg/m^2^, lenalidomide 25 mg, and dexamethasone 20 mg daily (*n* = 15); and VTD, bortezomib 1.3 mg/m^2^, thalidomide 50–100 mg, and dexamethasone 20 mg daily (*n* = 16) [24,25]. After therapy, 28 subjects received a single auto-HSCT, and 6 a tandem auto-HSCT. All patients received melphalan (140 to 200 mg/m^2^) as the conditioning regimen before transplantation. Post auto-HSCT therapies include daratumumab with or without lenalidomide and dexamethasone (*n* = 2), lenalidomide at 15 or 25 mg (*n* = 5), VTD or VRD (*n* = 2), or two or more treatments (*n* = 3). Nineteen patients received maintenance therapy. Seven subjects relapsed, and 9 died. Median follow-up was of 42.4 months (range, 10.4–103.9 months).

### 2.2. FLC Assay

Serum FLC were measured at diagnosis by nephelometric assay that detected the absolute free circulating kappa and lambda light chains. Diagnostic N Latex FLC kappa and N Latex FLC lambda kits (Siemens Healthcare Diagnostics, Milan, Italy) were used for FLC assessment as previously described [26]. Briefly, anti-FLC antibody-coated polystyrene particles were mixed with serum samples, and light dispersion was measured by turbidimetry using an Atellica^®^ CH Analyzer. Absorbance is proportional to FLC concentration, as free chains aggregate with coated polystyrene particles. Concentrations were calculated by interpolating unknown samples with a calibration curve, automatically prepared by the Atellica^®^ NEPH 630 system (Siemens Healthcare Diagnostics) by serially diluting the N FLC Standard SL with the N Diluent. Similarly, the system automatically diluted serum samples with N Diluent with a dilution factor of 1:100 for N Latex FLC kappa and 1:20 for N Latex FLC lambda. For a quality check, N FLC CONTROL SL1 and N FLC CONTROL SL2 controls were run after the calibration curve had been set, as per manufacturer instructions. For FLC kappa, the normal range was 6.7–22.4 mg/L, and for FLC lambda, it was 8.3–27.0 mg/L. Patients’ samples were analyzed within four hours from collection. After quantization, FLC ratio was automatically determined (normal range, 0.26–1.65). In our study, the standardized FLC (sFLC) ratio was used as MM biomarker and was calculated as the involved/uninvolved light-chain ratio.

### 2.3. Statistical Analysis

Data were analyzed using Prism (v.9.3.1; GraphPad software, La Jolla, CA, USA), and SPSS software. Unpaired two-tailed t-tests for two-group comparison, and one-way analysis of variance (ANOVA) for three-group comparison were performed. A log-rank (Mantel–Cox) test was employed for studying differences in overall survival (OS) and progression-free survival (PFS) between groups. The chi-squared test was used for categorical variable comparisons, while univariate and multivariate linear regression for the investigation of the association of FLC ratio values with other clinical and biological features. *p* < 0.05 was statistically significant.

## 3. Results

### 3.1. Clinical Characteristics of MM Patients with Abnormal sFLC Ratio

First, MM patients were divided in two groups based on sFLC ratio values (<1.65 or ≥1.65), and clinical characteristics at baseline and before and after auto-HSCT were compared to identify specific features of MM patients with increased sFLC ratio at diagnosis. The majority of patients (*n* = 29; 85.3%) displayed an abnormal sFLC ratio at diagnosis (median, 18.5; range, 2.4–3641). No significant differences in clinical feature distribution were observed between MM patients with increased sFLC ratio at diagnosis and those with normal values (Table 3). Interestingly, no variations were described in absolute FLC levels between patients with normal or abnormal FLC ratio for both involved (mean ± SD, 10.54 ± 2.6 mg/L vs. 1755.3 ± 4222.2 mg/L, normal vs. abnormal FLC ratio; *p* = 0.3684) and uninvolved chains (mean ± SD, 8.04 ± 2 mg/L vs. 12.06 ± 10.8 mg/L, normal vs. abnormal FLC ratio; *p* = 0.4170).

However, MM patients with light-chain M-protein type tended to have a significantly higher sFLC ratio compared to that of subjects with IgG/IgD/IgA M-protein types (mean ± SD, 1048 ± 1502 vs. 82.2 ± 217.9 vs. 9.65 ± 9.32, light chain vs. IgG vs. IgD/IgA; *p* = 0.0053). Similarly, patients with Stage 3 R-ISS tended to have higher sFLC ratio values compared to those subjects with Stage 1 or 2 MM at diagnosis (mean ± SD, 47.5 ± 11.1 vs. 99.4 ± 284.3 vs. 638.5 ± 1257, Stage 1 vs. 2 vs. 3; *p* = 0.0957) (Figure 1A,B). On the basis of the presence of renal failure, no differences were described for absolute involved FLC levels between groups (mean ± SD, 2948.4 ± 264.2 mg/L vs. 976.9 ± 2697.4 mg/L, presence vs. absence of renal failure; *p* = 0.2026). No variations in the type of first-line strategy, the presence of pretransplant renal failure, the type of pretransplant and best hematological responses, the rate of maintenance therapy, and clinical outcomes were described. No significant variations in OS and PFS were observed between groups either, as 5-year OS was 75% in MM patients with normal sFLC ratio at diagnosis, and 72.6% in those subjects with augmented sFLC ratio values (*p* = 0.8415); 5-year PFS was 60% and 51.5% in MM patients with normal or abnormal sFLC ratio at diagnosis, respectively (*p* = 0.7238). No significant variations were described in the percentage of BM pathological CD38^+^CD138^+^ plasma cells, even though subjects with increased sFLC ratio at diagnosis tended to have higher BM plasma cells compared to those with normal sFLC ratio values (mean ± SD, 13.5 ± 12.3% vs. 5.5 ± 6.3%, abnormal vs. normal sFLC ratio; *p* = 0.1751), and a slightly augmented ratio between BM plasma cells and residual hemopoiesis, defined as the sum of CD34^+^ and immature cells and CD19^+^ hematogones (mean ± SD, 1.46 ± 1.04 vs. 7.40 ± 9.48, normal vs. abnormal sFLC ratio; *p* = 0.1836) (Table 2) (Figure 1C,D).

No other relevant associations were observed with other flow cytometry data (Table 4), including BM frequency of lymphocytes (*p* = 0.9429), monocytes (*p* = 0.8151), granulocytes (*p* = 0.2600), and CD34^+^ cells (*p* = 0.7571). Neoplastic plasma cells positive for CD56, CD19, and CD45 were rarely found in our cohort (*n* = 1; 2.9%), while CD56^+^CD19^+^ or CD56^+^CD45^+^ clones were more frequently observed (*n* = 5, 14.7%; and *n* = 10, 29.4%; respectively) without significant variations in distribution between patients with abnormal sFLC ratio and normal values (*p* = 0.7174 and *p* = 0.6170, respectively).

### 3.2. Correlations with Clinical and Laboratory Features

Next, to explore possible correlations between sFLC ratio values, and clinical and laboratory features at diagnosis, Pearson analysis was performed (Figure 2). A significant association between sFLC ratio and β-2 microglobulin (*r* = 0.7549; *p* < 0.0001) and the percentage of BM CD34^+^ cells (*r* = 0.4186; *p* = 0.0468) was documented, while a negative correlation between sFLC ratio and glomerular filtration rate (GFR; *r* = −0.4484; *p* = 0.0078) was described.

No significant associations were described between sFLC ratio and age (*p* = 0.8513), albumin (*p* = 0.9638), lactate dehydrogenase (LDH; *p* = 0.6308), percentage of BM plasma cells by flow cytometry (*p* = 0.2550), lymphocytes (*p* = 0.3158), monocytes (*p* = 0.7967), granulocytes (*p* = 0.7502), immature cells (*p* = 0.5714), or hematogones (*p* = 0.3040). However, when patients were divided based on the presence of an already diagnosed renal failure, no differences in sFLC ratio values at diagnosis were described (mean ± SD, 465.2 ± 111.1 vs. 1202 ± 268.9, presence vs. absence of renal failure; *p* = 0.1672).

Lastly, univariate and multivariate linear regression analysis on sFLC ratio was performed (Table 5). The univariate model showed a significant association with β-2 microglobulin concentration (β-coefficient, 0.75; 95% confidential interval [CI], 99.9–192.7; *p* < 0.005) and light-chain MM type (β-coefficient, 0.61; 95% CI, 653.3–1827.3; *p* < 0.005); however, only β-2 microglobulin concentration was significantly associated with sFLC ratio in multivariate analysis (β-coefficient, 0.6; 95% CI, 67.7–166.3; *p* < 0.005).

## 4. Discussion

Serum sFLC assay detects excess light-chain immunoglobulins that are disproportionately produced in plasma cell disorders because of an abnormal κ/λ ratio [12,25,27]. Abnormal serum sFLC ratio values are related with MM prognosis and disease progression from MGUS to smoldering to high-risk MM [14,28]; its normalization after HSCT and before starting maintenance therapy is associated with lasting PFS and OS [29]. In this retrospective case series study, we evaluated the association of serum sFLC ratio with renal function and other laboratory findings to identify the additional prognostic impacts of this biomarker in MM progression, and renal impairment prediction and recovery. In our study, we investigated the distribution of the standardized FLC ratio (involved/uninvolved light chain) instead of absolute involved light-chain levels, showing that sFLC ratio was significantly associated with light-chain MM and β-2 microglobulin concentration, and inversely related with GFR, likely identifying early renal damage in MM patients.

Renal failure is present in 20 to 40% of MM patients at diagnosis and is mainly caused by accumulation of monoclonal light chains into renal tubules, and less frequently into glomeruli [30]. Hypercalcemia, dehydration, and nephrotoxic drugs are other contributing factors for the development of renal failure during MM [31]. The diagnosis of renal failure is based on serum creatinine levels (>2 mg/100 mL), urinary output, and GFR [32], and disease severity is assessed on the basis of estimated GFR values [33]. FLC > 800 mg/L at diagnosis is also linked to severe renal failure regardless of M-protein type, and quickly lowering FLC with antimyeloma therapies is associated with significant improvements in renal function [34]. In our case series, we first employed the standardized FLC ratio instead of serum FLC levels to normalize the results, rendering them comparable between studies and laboratories [35]. Bortezomib- and dexamethasone-based regimens with or without novel anti-CD38 treatments can rapidly reduce tumor burden and improve renal function, as almost one-quarter of MM patients with severe renal impairment can even stop dialysis [36,37,38]. The incidence of renal failure in newly diagnosed MM patients was 26.5%, similar to that already reported, and without differences based on the standardized FLC ratio. Interestingly, no variations were described based on absolute involved FLC ratio values between patients with renal failure and those with normal renal function. Moreover, when using the standardized ratio, we could better stratify patients, as no differences in involved FLC levels were described, even though FLC ratio was abnormal. An additional 25% of patients develop renal failure during the disease [30]; however, no additional patients in our cohort displayed a worsening in renal function during MM and before auto-HSCT. Those subjects had abnormal sFLC ratio at diagnosis (88.9%; *n* = 8/9) and were all treated with VTD or VRD regimens, supporting the safety of these protocols in MM patients with renal failure at diagnosis with a low rate of renal toxicities [39]. Recovery of renal function is expected with novel therapeutic treatments, especially for those patients with severe renal impairment or renal replacement with dialysis [36]. In our cohort, no patients had severe renal failure or were under dialysis; therefore, renal recovery was not expected, while the absence of worsening of previously present impairment of the development of treatment-related failure suggested the safety of bortezomib- and dexamethasone-based regimens in both reducing tumor burden and the risk or renal impairment.

β-2 microglobulin, a member of the β globulin family, is associated with human leukocyte antigen I on the cell surface, and is involved in antigen presentation, mucosal immunity, and tumor surveillance [40]. This globulin is used as a marker of tumor burden in lymphoid malignancies and MM, as increased serum levels and high LDH are related with poor survival and increased risk of disease progression [41,42]. Serum β-2 microglobulin and LDH levels have been incorporated in the International Staging System (ISS) for MM and in the Revised-ISS (R-ISS) together with serum albumin [42,43,44]. The main limitation of these biomarkers in risk stratification of plasma cell disorders is that they are not specific of MM; for example, serum β-2 microglobulin levels are affected by renal function and are a biomarker of kidney failure, as renal filtration can be estimated based on β-2 microglobulin and creatinine-related eGFR [40]. In our study, serum sFLC ratio positively correlated with β-2 microglobulin and negatively with GFR, while no associations were described for LDH and albumins. Therefore, correlations between sFLC ratio, and β-2 microglobulin and GFR suggest that serum free κ/λ chain ratio could also be a biomarker of kidney function and might predict renal failure early in plasma cell disorders. Our cohort only comprised MM patients; thus, a prospective study on larger populations with also MGUS and smoldering MM should be included to evaluate the prognostic significance of serum sFLC ratio in the early identification of kidney disfunction in those patients.

FLC ratio can be normal, especially in pre-myeloma stages with low risk of progression, in younger patients, or in those subjects with concomitant increased of uninvolved light-chain levels [45,46,47]; however, this condition is not frequent in active MM, as reported in our case series (only 18% of patients, mostly with concomitant increased of uninvolved FLC), also in accordance with previously published studies [47,48]. Conversely, FCL ratio is abnormal in almost all non-secretory myeloma with a sensitivity close to 100% [16,17,18,19]. In our case series, we confirmed the high diagnostic potential of sFLC ratio for light-chain MM identification, and we also showed that the standardized values were highly correlated with renal failure, especially in light-chain MM.

M protein concentration is another key marker of diagnosis and follow-up of plasma cell dyscrasia [42]. Negative serum and urinary immunofixation is still one of the main criterium to define treatment response in MM patients according to the IMWG consensus criteria [49]; however, when R-ISS is combined with molecular alterations, better risk stratification is achieved supporting the clinical utility of next-generation sequencing and highlighting the need of a multidisciplinary approach to better define the prognosis of hematological malignancies [12,50]. Moreover, M proteins can rarely be absent in non-secretory myeloma, and additional disease biomarkers are required to monitor patients during treatment and follow-up [51,52]. In our study, only a slight correlation was described between sFLC ratio values and M-protein concentration, as these two biomarkers together reflect disease burden, while sFLC ratio can better define prognosis because increased sFLC ratio alone can predict early and at shorter time to second line-treatment, and higher risk of MM progression and mortality [42,51,52].

Cytogenetic abnormalities detected by fluorescence in situ hybridization (FISH) are included in the R-ISS for MM and have a major prognostic impact, as high-risk cytogenetics, such as the presence of del(17p), identify patients with poor prognosis regardless of β-2 microglobulin and LDH levels [42,53]. In our case series, no significant associations were found between abnormal sFLC ratio values and the presence of cytogenetic abnormalities, including del(17p), while patients with high R-ISS stages tended to have increased sFLC ratio at diagnosis compared to early-stage MM, likely because of concomitant high β-2 microglobulin.

Multiparametric flow cytometry is the standard method to determine measurable minimal residual disease with a sensitivity of one clonal plasma cell in 10^5^ cells [54], and the ability to identify certain plasma cell phenotypes with worse outcomes, such as CD117^+^ plasma cell precursors or CD38^low^CD81^+^CD117^−^ cells [55,56,57]. When BM plasma cell frequency is combined with serum sFLC ratio and M protein concentration, better risk stratification and risk of progression from smoldering to MM are reported [11,12,16,58]. In our case series, sFLC ratio values tended to increase with the percentage of BM CD34^+^ cells and neoplastic CD38^+^CD138^+^ plasma cells, and a slightly augmented ratio between BM plasma cells and residual hemopoiesis, defined as the sum of CD34^+^ and immature cells and CD19^+^ hematogones. Therefore, our results suggest that sFLC ratio might be a surrogate of the BM hemopoiesis status and disease burden measured by multiparametric flow cytometry.

The limitations of our study are: the retrospective nature and the small number of enrolled patients; the absence of MGUS or smoldering MM patients to investigate the prognostic role of sFLC ratio in disease progression and renal failure development under novel therapeutic treatments; the absence of MM under dialysis treatment for the investigation of renal recovery after sFLC normalization; and the study was not powered for subgroup analysis by other risk factors.

## 5. Conclusions

In conclusion, MM patients can develop renal failure later in their clinical course, and the biological causes behind this late onset are still poorly understood. Hypercalcemia is one of the possible reasons, and the monitoring of serum calcium level might help in the clinical management of MM patients; however, serum calcium could be affected by MM treatment and not considered to be a specific and sensitive biomarker of renal failure. Here, we showed that abnormal standardized FLC ratio in newly diagnosed MM patients might early identify renal disfunction and patients with late-stage disorder, supporting the use of this normalized biomarker as a complementary assay in the initial evaluation of plasma cell dyscrasia and in the follow-up of patients.

## Figures and Tables

**Figure 1 biomedicines-10-01657-f001:**
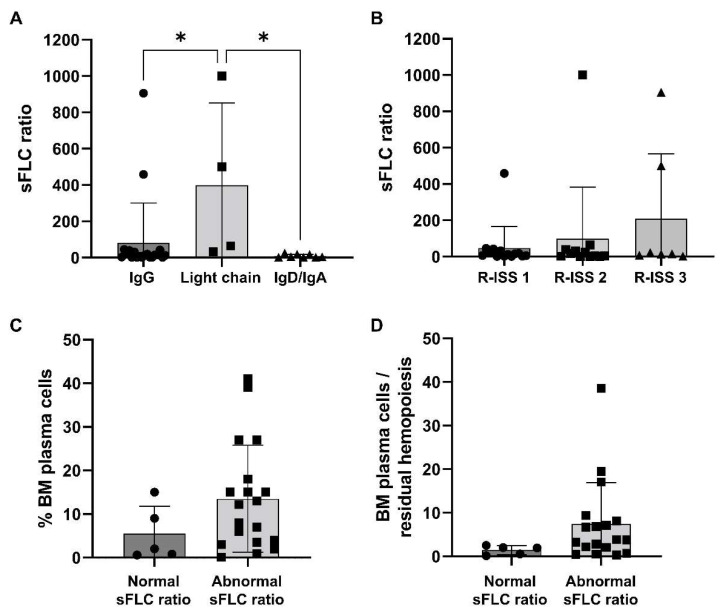
Serum standardized free light-chain (sFLC) ratio distribution. (**A**) Patients were divided based on multiple myeloma (MM) types, such as immunoglobulin (Ig) G, IgD, or IgA, and light-chain types and sFLC ratio values were compared among groups. (**B**) Similarly, sFLC ratio values at diagnosis were compared based on Revised International Staging System (R-ISS) stages. According to the normal range of the sFLC ratio, patients were divided based on normal or abnormal values at diagnosis and (**C**) percentage of bone marrow (BM) plasma cells or (**D**) the ratio between BM plasma cells and residual hemopoiesis were compared among groups. Data are shown as mean + SD. * *p* < 0.05.

**Figure 2 biomedicines-10-01657-f002:**
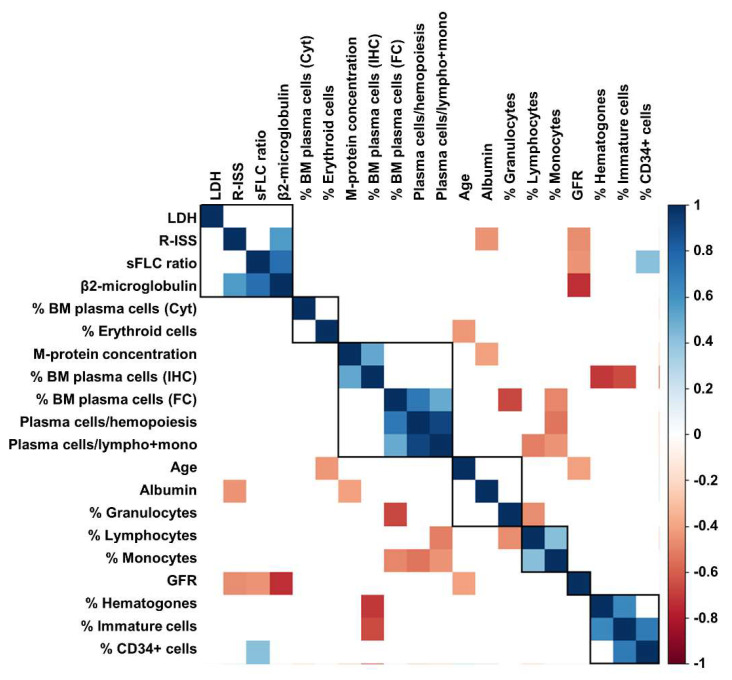
Correlation analysis between clinical and laboratory markers. Correlation analysis was performed with Pearson analysis among serum standardized free light-chain (sFLC) ratio, lactate dehydrogenase (LDH), Revised International Staging System (R-ISS) stages, bone marrow (BM) plasma cells detected by morphology (Cyt), immunohistochemistry (IHC) or flow cytometry (FC), percent of BM populations detected by FC, laboratory, and clinical findings, such as glomerular filtration rate (GFR).

**Table 1 biomedicines-10-01657-t001:** Clinical characteristics at baseline.

Characteristics	*n* = 34
Median age, years (range)	60.5 (39–73)
M/F	18/16
ECOG Performance status	
0	18 (53%)
1	13 (38.2%)
2	3 (8.8%)
M-protein type	
IgG	20 (58.8%)
Light chain	5 (14.7%)
IgD	1 (2.9%)
IgA	7 (20.6%)
Not specified	1 (2.9%)
Type of free light chains	
κ/λ	17 (50%)/17 (50%)
R-ISS stage	
1	14 (41.2%)
2	12 (35.3%)
3	8 (23.5%)
Cytogenetic abnormalities	
del(17p)	4 (11.8%)
Normal	15 (44.1%)
Others	6 (17.6%)
Not performed	9 (26.5%)
Median BM plasma cells, % (range)	46 (10–80)
Median M-protein, g/dL (range)	2.68 (0–5.9)
Median involved chain, mg/L (range)	128 (7.9–19,300)
Median uninvolved chain, mg/L (range)	9.1 (1.8–54.6)
Median FLC ratio	16.15 (1–3641)
Increased FLC ratio	28/34 (82%)
Presence of urinary Bence Jones protein	20/34 (58.8%)
Median β2-microglobulin, mg/dL (range)	3.3 (1.4–19)
Median LDH, U/L (range)	395 (217–901)
Median albumin, g/dL (range)	3.35 (2–4.9)

Abbreviations. Ig, immunoglobulin; R-ISS, revised international staging system; BM, bone marrow; FLC, free light chain; LDH, lactate dehydrogenase.

**Table 2 biomedicines-10-01657-t002:** Treatments and outcomes.

	*n* = 34
First line therapy	
PAD	2 (5.9%)
VD	1 (2.9%)
VRD	15 (44.1%)
VTD	16 (47.1%)
Auto-HSCT	
Single/tandem	28 (82%)/6 (18%)
Post-HSCT treatment	12/34 (35.3%)
Daratumumab + RD	2 (16.7%)
Lenalidomide	5 (41.6%)
VTD/VRD	2 (16.7%)
>2 lines	3 (25%)
Maintenance treatment	19/34 (55.9%)
Relapse	7/34 (20.6%)
Best response	
SD	2 (5.9%)
PR	2 (5.9%)
VgPR	9 (26.5%)
CR/sCR	21 (61.7%)
Deaths	9 (26.5%)
Median follow-up, months (range)	42.4 (10.4–103.9)

Abbreviations. PAD, bortezomib, dexamethasone, and doxorubicin; VD, bortezomib and dexamethasone; VRD, bortezomib, lenalidomide, and dexamethasone; VTD, bortezomib, thalidomide, and dexamethasone; HSCT, hematopoietic stem cell transplantation; auto-HSCT, autologous HSCT; RD, lenalidomide and dexamethasone; SD, stable disease; PR, partial remission; VgPR, very good partial remission; CR, complete remission; sCR, stringent CR.

**Table 3 biomedicines-10-01657-t003:** Clinical characteristics based on sFLC ratio values.

Characteristics	Normal sFLC Ratio	Abnormal sFLC Ratio	*p* Value
	*n* = 5	*n* = 29	
Median age, years (range)	57 (52–65)	62 (39–73)	0.8121
M/F	2/3	16/13	0.6481
ECOG Performance Status			0.3894
0–1	4	27	
2	1	2	
Presence of comorbidities	1/5	17/29	0.1642
M-protein type			0.6253
IgG/Others	4/1	16/13	
Type of free light chains			>0.9999
κ/λ	2/3	15/14	
R-ISS stage			0.3086
1–2	5	21	
3	0	8	
Cytogenetic abnormalities			0.2500
Normal/others	3/0	12/10	
Not performed	2	7	
Presence of del(17p)/absence of del(17p)	0/3	4/22	>0.9999
Not performed	2	3	
Median M-protein, g/dL (range)	3.12 (2.75–5.23)	1.89 (0–5.9)	0.1024
Median involved chain, mg/L (range)	9.8 (7.9–14.9)	163 (15.6–19, 300)	0.3684
Median uninvolved chain, mg/L (range)	8 (6.1–10.9)	9.4 (1.8–54.6)	0.4170
Median FLC ratio	1.3 (1–1.5)	18.5 (2.4–3641)	0.4598
Presence of urinary Bence Jones protein	1/5	19/29	0.1349
Median β2-microglobulin, mg/dL (range)	3.3 (2.92–3.8)	3.3 (1.4–19)	0.9874
Abnormal/normal β2-microglobulin	5/0	23/6	0.5585
Median LDH, U/L (range)	402 (364–901)	394 (217–755)	0.1959
Median albumin, g/dL (range)	2.9 (2.7–3.8)	3.4 (2–4.9)	0.2878
Median GFR, mL/min (range)	112 (55–115)	90 (4–123)	0.2214
Abnormal GFR	1/5	7/29	>0.9999
Presence of renal failure	1/5	8/29	>0.9999
Presence of osteolytic lesions	3/5	17/29	0.6343
More than 2 osteolytic lesions	1/5	13/29	>0.9999
Presence of hypercalcemia	1/5	2/29	0.3894
Presence of anemia	3/5	20/29	>0.9999
Pre-HSCT renal failure	1/5	7/29	>0.9999
Pre-HSCT status			>0.9999 >0.9999
CR	4	19	
SD/PR	1	10	
Auto-HSCT			
Single/tandem	4/1	24/5	
Maintenance treatment	2/5	17/29	0.6343
Relapse	2/5	5/29	0.2684
Best response			0.1317
SD	0	2	
PR	0	2	
VgPR	0	9	
CR/sCR	5	16	
Deaths	1/5	8/29	>0.9999
Median OS, months (range)	68.9 (23.9–73.9)	41.9 (10.4–103.9)	0.8415
Median PFS, months (range)	67 (20–73.9)	40.6 (6.9–103.9)	0.7238

Abbreviations. Ig, immunoglobulin; R-ISS, revised international staging system; BM, bone marrow; FLC, free light chain; LDH, lactate dehydrogenase; eGFR, estimated glomerular filtration rate; PAD, bortezomib, dexamethasone, and doxorubicin; VD, bortezomib and dexamethasone; VRD, bortezomib, lenalidomide, and dexamethasone; VTD, bortezomib, thalidomide, and dexamethasone; HSCT, hematopoietic stem cell transplantation; auto-HSCT, autologous HSCT; RD, lenalidomide and dexamethasone; SD, stable disease; PR, partial remission; VgPR, very good partial remission; CR, complete remission; sCR, stringent CR.

**Table 4 biomedicines-10-01657-t004:** Flow cytometry features based on sFLC ratio values.

Characteristics	Normal sFLC Ratio *n* = 5	Abnormal sFLC Ratio *n* = 29	*p* Value
Median CD38^+^CD138^+^ plasma cells, % (range)	2 (0.6–15)	12.2 (0.1–41)	0.1751
Median lymphocytes, % (range)	16 (9–35)	18 (1–35)	0.9429
Median monocytes, % (range)	4 (3–7)	5 (1–11)	0.8151
Median granulocytes, % (range)	68 (54–79)	64 (28–72)	0.2600
Median CD34^+^ cells, % (range)	0.6 (0–1.1)	0.6 (0.2–1.4)	0.7571
Plasma cells/residual hemopoiesis	1.43 (0.2–1.95)	2.14 (0.3–38.6)	0.1812
CD56^+^ plasma cells	5/5	25/27	>0.9999
CD19^+^ plasma cells	2/5	4/27	0.2779
CD45^+^ plasma cells	2/5	10/27	>0.9999
CD56^+^CD19^+^CD45^+^ plasma cells	1/5	0/27	0.1562
Double positive plasma cells	2/5	13/27	>0.9999

Abbreviations. PAD, bortezomib, dexamethasone, and doxorubicin; VD, bortezomib and dexamethasone; VRD, bortezomib, lenalidomide, and dexamethasone; VTD.

**Table 5 biomedicines-10-01657-t005:** Univariate and multivariate linear regression model.

Univariate			
	β Coefficient	95% CI	*p* Value
β-2 microglobulin	0.75	99.9–192.7	<0.005
Gender (male)	−0.19	−723.5/220.1	0.28
Age	−0.05	−36/26.1	0.75
MM type (light chain)	0.61	653.5/1827.3	<0.005
M-protein at diagnosis	−0.33	−238.3/2	0.54
Light-chain type (lambda)	0.17	−248.4/695.4	0.34
LDH	0.07	−1.2/1.8	0.66
Albumin	0.004	−394.5/404.2	0.98
% BM plasma cells	−0.09	−30.9/18.8	0.62
Extramedullary disease (yes)	−0.05	−671.4/497.9	0.76
**Multivariate**			
	**β coefficient**	**95% CI**	***p* value**
MM type (light chain)	0.28	−2.5/1128.1	0.05
M-protein at diagnosis	−0.11	−131.3/50.6	0.37
β-2 microglobulin	0.6	67.7/166.3	<0.005

Abbreviations. FLC, free light chain; MM, multiple myeloma; LDH, lactate dehydrogenase; BM, bone marrow.

## Data Availability

Data are available upon request by the authors.

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
