# Peer review of "Serum Free Light-Chain Ratio at Diagnosis Is Associated with Early Renal Damage in Multiple Myeloma: A Case Series Real-World Study"

_biomedicines, 2022, doi:10.3390/biomedicines10071657_

Round 1

Reviewer 1 Report

Reviewer’s comment:

Biomarker detection is urgent need for the multiple myeloma (MM) progression and plasma cell disorder. There are some questions need to be answer.

1. Table 2, In the Deaths column, Why the number of patients are 31?

2. MethodS:

  (1) 2.2 FLC assay should be descript more detail. Because it is the highlight of the article.       

  (2) The FLC in the serum will be degraded or reabsorption during 2-4 hours. When and how did you take the serum and detection?

3. Table 3

  (1) In the cytogenetic abnormalities and 17p deletion column, the number is not equal to the N value

  (2) M-protein type in the abnormal sFLC ratio column is not equal to 29.

  (3) In the abnormal b2-microglobulin, the range is between 1.4-1.9, but the average is 3.3 mg/dL. The average is out of the range.

4. Figure 2

  (1) The authors said that “a slight correlation was present between sFLC ratio and M-protein concentration or R-ISS stage”. But in the figure, R-ISS stage was observed correlation with b2-microglobulin, albumin, GFR not sFLC. The same as M protein concentration is correlated with % BM plasma cells (FC) and albumin.

Author Response

Reviewer 1

Biomarker detection is urgent need for the multiple myeloma (MM) progression and plasma cell disorder. There are some questions need to be answer.

Comment 1. Table 2, In the Deaths column, Why the number of patients are 31?

Response to Comment 1. We thank the Reviewer for this comment. We apologize for this typo. The number is 34 and the percentage was correct (26.5%).

Comment 2. FLC assay should be descript more detail. Because it is the highlight of the article.

Response to Comment 2. We thank the Reviewer for this comment. We have improved the 2.2 FLC assay section as follows.

On page 4, lines 109-126, the following text was added “Serum FLC were measured at diagnosis by nephelometric assay that detected the absolute free circulating kappa and lambda light chains. Diagnostic N Latex FLC kappa and N Latex FLC lambda kits (Siemens Healthcare Diagnostics, Milan, Italy) were used for FLC assessment as previously described [26]. Briefly, anti-FLC anti-body-coated polystyrene particles were mixed with serum samples, and light dispersion was measured by turbidimetry using an Atellica® CH Analyzer. Absorbance is proportional to FLC concentration, as free chains aggregate with coated polystyrene particles. Concentrations were calculated by interpolating unknown samples with a calibration curve, automatically prepared by the Atellica® NEPH 630 system (Siemens Healthcare Diagnostics) by serially diluting the N FLC Standard SL with the N Diluent. Similarly, the system automatically diluted serum samples with N Diluent with a dilution factor of 1:100 for N Latex FLC kappa and 1:20 for N Latex FLC lambda. For quality check, N FLC CONTROL SL1 and N FLC CONTROL SL2 controls were run after calibration curve was set, as per manufacturers’ instructions. For FLC kappa, normal range was 6.7–22.4 mg/L, and for FLC lambda was 8.3–27.0 mg/L. Patients’ samples were analyzed within four hours from collection. After quantization, FLC ratio was automatically determined (normal range, 0.26-1.65). In our study, the standardized FLC (sFLC) ratio was used as MM biomarker and was calculated as the in-volved/uninvolved light chain ratio.”

Comment 3. The FLC in the serum will be degraded or reabsorption during 2-4 hours. When and how did you take the serum and detection?

Response to Comment 3. Samples are processed within 4 hours from collection, as also added in 2.2. FLC assay section.

Comment 4. Table 3. In the cytogenetic abnormalities and 17p deletion column, the number is not equal to the N value. M-protein type in the abnormal sFLC ratio column is not equal to 29. In the abnormal b2-microglobulin, the range is between 1.4-1.9, but the average is 3.3 mg/dL. The average is out of the range.

Response to Comment 4. We apologize for this. We have added additional lines for “not performed” analysis, and the range is 1.4-19 mg/dL.

Comment 5. Figure 2. The authors said that “a slight correlation was present between sFLC ratio and M-protein concentration or R-ISS stage”. But in the figure, R-ISS stage was observed correlation with b2-microglobulin, albumin, GFR not sFLC. The same as M protein concentration is correlated with % BM plasma cells (FC) and albumin.

Response to Comment 5. We thank the Reviewer for the comment, and we have removed that sentence because it was misleading.

Reviewer 2 Report

Novellis, et al. reported the clinical relevance of the serum free light chain (FLC) ratio in 34 patients with transplant-eligible multiple myeloma. The authors investigated the association between the FLC ratio and clinical parameters, and found that the FLC ratio was significantly associated with micromolecular type of myeloma, b2-microglobulin, and the glomerular filtration rate. Based on these data, they concluded that the FLC ratio is useful for evaluation of renal function in patients with multiple myeloma, especially, micromolecular or non-secretory type. However, the strong relationship between FLC and renal dysfunction has been already reported, for example, Yadev, et al. BMC Nephrol 19:178, 2018. Moreover, there are several concerns regarding the study that the authors need to clarify.

1. The significance and indication of FLC ratio has been already established by the IMWG guidelines (Dispenzieri, et al. Leukemia 23:215, 2009). It is recommended to cite such literature and describe an overview on FLC in Introduction.

2. What is micromolecular type of myeloma? Is that light chain only or Bence-Jones type?

3. Since the FLC ratio is influenced by the value of the non-involved light chain, the absolute amount of the involved FLC is considered to be important. What is the result by the amount of FLC?

4. In Fig. 1A and B, the scale is too large due to the case showing an abnormally high value, and it is difficult to understand the distribution of most of the cases.

5. In Fig. 1C, there are some cases where the proportion of myeloma cells is less than 10%, which does not meet the diagnostic criteria of multiple myeloma.

6. Since the heavy chain to be bound is missing in the light chain only type (micromolecular?) of myeloma, it seems obvious that the serum FLC increases, but what is the reason of emphasizing it in conclusion?

7. The authors need to add a discussion about the mechanism by which the FLC ratio is normal.

8. Recovery of renal damage is expected with new treatment modalities, and it is recommended to consider it in the view of FLC ratio

9. It is difficult to understand the discussion of the relationship between oncogenes and immunophenotype of myeloma cells and the FLC ratio.

Author Response

Novellis, et al. reported the clinical relevance of the serum free light chain (FLC) ratio in 34 patients with transplant-eligible multiple myeloma. The authors investigated the association between the FLC ratio and clinical parameters, and found that the FLC ratio was significantly associated with micromolecular type of myeloma, b2-microglobulin, and the glomerular filtration rate. Based on these data, they concluded that the FLC ratio is useful for evaluation of renal function in patients with multiple myeloma, especially, micromolecular or non-secretory type. However, the strong relationship between FLC and renal dysfunction has been already reported, for example, Yadev, et al. BMC Nephrol 19:178, 2018. Moreover, there are several concerns regarding the study that the authors need to clarify.

Response to General Comment. We thank the Reviewer for this feedback. We agree with the Reviewer; however, we have focused more on the standardized FLC ratio rather than absolute involved FLC levels.  

On pages 9-10, lines 256-270, the following text was added “FLC > 800 mg/L at diagnosis is also linked to severe renal failure regardless M-protein type, and quickly lowering FLC with anti-myeloma therapies is associated with significant improvements of renal function [34]. In our case series, first, we employed the standardized FLC ratio instead of serum FLC levels to normalize results making them comparable between studies and laboratories [35]. Bortezomib- and dexamethasone- based regimens, with or without novel anti-CD38 treatments, can rapidly reduce tumor burden and improve renal function, as almost a quarter of MM patients with severe renal impairment can even stop dialysis [36-38]. We showed that the incidence of renal failure in newly diagnosed MM patients was 26.5%, similar to that already re-ported, and without differences based on standardized FLC ratio. Interestingly, no variations were described based on absolute involved FLC ratio values between patients with renal failure and those with normal renal function. Moreover, when using the standardized ratio, we could better stratify patients, as no differences in involved FLC levels were described even though FLC ratio was abnormal.”

Comment 1. The significance and indication of FLC ratio has been already established by the IMWG guidelines (Dispenzieri, et al. Leukemia 23:215, 2009). It is recommended to cite such literature and describe an overview on FLC in Introduction.

Response to Comment 1. We thank the Reviewer for this suggestion. On page 2, lines 54-59, the following text was added “Indeed, sFLC has been a breakthrough biomarker of MM with a sensitivity close to 100% for non-secretory myeloma, as it FLC ratio is always abnormal even in the absence of M-protein thus making this marker an optimal tool for monitoring oligo- or non-secretory plasma cell disorders [16-19]. Moreover, sFLC is used for screening together with protein electrophoresis, immunofixation, and 24-h urine studies [17].

The following references were added.

  1. Heaney, J.L.J.; Richter, A.; Bowcock, S.; Pratt, G.; Child, J.A.; Jackson, G.; Morgan, G.; Turesson, I.; Drayson, M.T. Excluding myeloma diagnosis using revised thresholds for serum free light chain ratios and M-protein levels. Haematologica. 2020, 105, e169-e171.
  2. Dispenzieri, A.; Kyle, R.; Merlini, G.; Miguel, J.S.; Ludwig, H.; Hajek, R.; Palumbo, A.; Jagannath, S.; Blade, J.; Lonial, S.; Di-mopoulos, M.; Comenzo, R.; Einsele, H.; Barlogie, B.; Anderson, K.; Gertz, M.; Harousseau, J.L.; Attal, M.; Tosi, P.; Sonneveld, P.; Boccadoro, M.; Morgan, G.; Richardson, P.; Sezer, O.; Mateos, M.V.; Cavo, M.; Joshua, D.; Turesson, I.; Chen, W.; Shimizu, K.; Powles, R.; Rajkumar, S.V.; Durie, B.G.; International Myeloma Working Group. International Myeloma Working Group guidelines for serum-free light chain analysis in multiple myeloma and related disorders. Leukemia. 2009, 23, 215-224.
  3. Milani, P.; Palladini, G.; Merlini, G. Serum-free light-chain analysis in diagnosis and management of multiple myeloma and related conditions. Scand J Clin Lab Invest Suppl. 2016, 245, S113-118.
  4. Markovic, U.; Leotta, V.; Tibullo, D.; Giubbolini, R.; Romano, A.; Del Fabro, V.; Parrinello, N.L.; Cannizzaro, M.T.; Di Rai-mondo, F.; Conticello, C. Serum free light chains and multiple myeloma: Is it time to extend their application? Clin Case Rep. 2020, 8, 617-624.

Comment 2. What is micromolecular type of myeloma? Is that light chain only or Bence-Jones type?

Response to Comment 2. We thank the Reviewer for this comment. We have changed “micromolecular” to “light chain MM”.

Comment 3. Since the FLC ratio is influenced by the value of the non-involved light chain, the absolute amount of the involved FLC is considered to be important. What is the result by the amount of FLC?

Response to Comment 3. We thank the Reviewer for this comment, and we have added these results in Table 3 with appropriate statistical analysis between groups.

On page 5, lines 145-149, the following text was added “Interestingly, no variations were described in absolute FLC levels between patients with normal or abnormal FLC ratio for both involved (mean+SD, 10.54+2.6 mg/L vs 1755.3+4,222.2 mg/L, normal vs abnormal FLC ratio; P = 0.3684) and uninvolved chains (mean+SD, 8.04+2 mg/L vs 12.06+10.8 mg/L, normal vs abnormal FLC ratio; P = 0.4170).”

On page 5, lines 165-168, the following text was added “Based on the presence of renal failure, no differences were described for absolute in-volved FLC levels between groups (mean+SD, 2948.4+264.2 mg/L vs 976.9+2697.4 mg/L, presence vs absence of renal failure; P = 0.2026).”

Comment 4. In Fig. 1A and B, the scale is too large due to the case showing an abnormally high value, and it is difficult to understand the distribution of most of the cases.

Response to Comment 4. We have removed that patient with abnormally high sFLC from the figure (not from statistical analysis) for clarity as requested.

Comment 5. In Fig. 1C, there are some cases where the proportion of myeloma cells is less than 10%, which does not meet the diagnostic criteria of multiple myeloma.

Response to Comment 5. We agree with the Reviewer that the percentage of plasma cells is less than 10% in some cases; however, these are flow cytometry data from BM aspirate specimens that are known to underestimate plasma cell frequency.

Comment 6. Since the heavy chain to be bound is missing in the light chain only type (micromolecular?) of myeloma, it seems obvious that the serum FLC increases, but what is the reason of emphasizing it in conclusion?

Response to Comment 6. We thank the Reviewer for this comment, and we have added some discussion for this point.

On page 10, lines 301-309, the following text was added “FLC ratio can be normal, especially in pre-myeloma stages with low risk of progression, younger patients, or in those subjects with concomitant increased of uninvolved light chain levels [45-48]; however, this condition is not frequent in active MM, as reported in our case series (only 18% of patients, mostly with concomitant increased of uninvolved FLC), also in accordance with previously published studies [48-49]. Conversely, FCL ratio is abnormal in almost all non-secretory myeloma with a sensitivity close to 100% [16-19]. In our case series, we confirmed the high diagnostic potential of sFLC ratio for light chain MM identification, and we also showed that standardized values were highly correlated with renal failure especially in light chain MM.”

Comment 7. The authors need to add a discussion about the mechanism by which the FLC ratio is normal.

Response to Comment 7. Please refer to Response to Comment 6.

Comment 8. Recovery of renal damage is expected with new treatment modalities, and it is recommended to consider it in the view of FLC ratio.

Response to Comment 8. We thank the Reviewer for raising this comment, and the following text was added on page 10, lines 275-281 “Recovery of renal function is expected with novel therapeutic treatments, especially for those patients with severe renal impairment or renal replacement with dialysis [36]. In our cohort, no patients had severe renal failure or were under dialysis; there-fore, renal recovery was not expected while absence of worsening of previously present impairment of development of treatment-related failure suggested safety of bortezomib- and dexamethasone-based regimens in both reducing tumor burden and the risk or renal impairment.”

Comment 9. It is difficult to understand the discussion of the relationship between oncogenes and immunophenotype of myeloma cells and the FLC ratio.

Response to Comment 9. We agree with the Reviewer as that point was out of the context; therefore, we have removed that sentence.

Round 2

Reviewer 1 Report

It is ok. Good Luck!!

Reviewer 2 Report

The authors revised the manuscript appropriately according to the reviewers' comments.

This manuscript is a resubmission of an earlier submission. The following is a list of the peer review reports and author responses from that submission.